# Diversity-Driven View Subset Selection for Indoor Novel View Synthesis

**Zehao Wang**                                          *zehao.wang@esat.kuleuven.be*
*ESAT - Processing Speech and Images*
*KU Leuven*

**Han Zhou**                                            *han.zhou@esat.kuleuven.be*
*ESAT - Processing Speech and Images*
*KU Leuven*

**Matthew B. Blaschko**                                 *matthew.blaschko@esat.kuleuven.be*
*ESAT - Processing Speech and Images*
*KU Leuven*

**Tinne Tuytelaars**                                    *tinne.tuytelaars@esat.kuleuven.be*
*ESAT - Processing Speech and Images*
*KU Leuven*

**Minye Wu**                                            *minye.wu@esat.kuleuven.be*
*ESAT - Processing Speech and Images*
*KU Leuven*

**Reviewed on OpenReview:** *https://openreview.net/forum?id=F42CRfcp3D*

## Abstract

Novel view synthesis of indoor scenes can be achieved by capturing a monocular video sequence of the environment. However, redundant information caused by artificial movements in the input video data reduces the efficiency of scene modeling. To address this, we formulate the problem as a combinatorial optimization task for view subset selection. In this work, we propose a novel subset selection framework that integrates a comprehensive diversity-based measurement with well-designed utility functions. We provide a theoretical analysis of these utility functions and validate their effectiveness through extensive experiments. Furthermore, we introduce INDOORTRAJ, a novel dataset designed for indoor novel view synthesis, featuring complex and extended trajectories that simulate intricate human behaviors. Experiments on INDOORTRAJ show that our framework consistently outperforms baseline strategies while using only 5–20% of the data, highlighting its remarkable efficiency and effectiveness. The code is available at: `https://github.com/zehao-wang/IndoorTraj`.

## 1 Introduction

Novel view synthesis using a monocular RGB camera in indoor environments holds significant practical value for real-world applications. This technology allows users to reconstruct indoor scenes from self-captured video data, making it particularly beneficial for VR/AR applications, such as virtual tours. However, these videos often contain lengthy sequences with redundant information caused by complex camera movements resulting from human actions. This redundancy makes scene modeling less efficient, leading to longer processing times. It is crucial to develop methods that efficiently select a subset of frames within memory constraints. These target methods aim to maintain novel view synthesis performance that is comparable to using the full frame set, while significantly reducing memory and computation demands.

In this paper, we formulate this problem as a combinatorial optimization task that aims to maximize novel view synthesis quality with a fixed subset size $K$. The function $z(\mathbb{S})$ serves as our utility function to measure the novel view synthesis quality, where $\mathbb{S}$ is a subset of the full set $\mathbb{D}$. This subset selection problem is expressed as follows:

$$\max_{\mathbb{S} \subseteq \mathbb{D}} \left\{ z(\mathbb{S}) : |\mathbb{S}| \leq K \right\}, \tag{1}$$

The quality of novel view synthesis is typically evaluated using the Peak Signal-to-Noise Ratio (PSNR) and related metrics applied to synthesized 2D images. However, computing these metrics requires expensive reconstruction and rendering for each chosen subset $\mathbb{S}$. Moreover, selecting a subset of size $K$ that maximizes performance among all possible combinations poses an NP-hard challenge. All of these computational complexity make it impossible to solve this subset selection problem in practice.

From the literature, several measures have been identified as useful for assessing reconstruction quality. Warping consistency (Ye et al., 2024) and reconstruction uncertainty (Wu et al., 2015; Goli et al., 2024; Pan et al., 2022) emphasize per-camera importance and are effective for Next-Best-View (NBV) selection tasks. The calculation of these measures no longer requires completing the full reconstruction process; they can be computed during several iterations of the reconstruction, significantly reducing computational intensity. Nevertheless, they still require substantial training and inference efforts, making them less ideal for the efficiency requirements of subset selection problems. Another category of measures, focusing on view diversity, is commonly used in SLAM and neural SLAM tasks (Raúl Mur-Artal & Tardós, 2015; Sucar et al., 2021; Yang et al., 2022; Zhu et al., 2022). These diversity-based methods focus on geometric criteria such as distance and frustum overlap, but fail to capture angle variation and scene semantics comprehensively. Building on these insights, we propose an efficient multifactor distance measure that combines pairwise distances across 3D, angular, and semantic spaces. This measure jointly promotes spatial coverage, directional diversity, and content variation, ensuring the selected views encompass a wide physical area, varied perspectives, and diverse scene content to better support view selection and scene reconstruction.

Given the distance measure and objective of our task, we define two utility functions $z(\mathbb{S})$ and apply an efficient greedy algorithm to maximize $z(\mathbb{S})$ by iteratively optimizing its marginal contribution. The first utility function is adopted from determinantal point process (DPP) (Kulesza et al., 2012), a probabilistic model known for its effectiveness in selecting diverse subsets. The second is inducted from diversity related heuristics used in recommender system (Carbonell & Goldstein, 1998) and NBV (Xiao et al., 2024) selection. While these heuristics have shown good empirical performance in other tasks, their effectiveness in the view subset selection problem has not been thoroughly explored. Additionally, we analyze a coverage-based utility function derived from Kopanas & Drettakis (2023). We reformulate the first two utility functions using our distance measure and assess the effectiveness of all three both theoretically and empirically.

Existing indoor scene datasets often feature fixed camera movement speeds or relatively short and simple trajectories, such as those in Straub et al. (2019); Zhu et al. (2023). To enable more comprehensive experiments that better simulate complex human capture behaviors, we introduce a new dataset called INDOORTRAJ. Experimental results on INDOORTRAJ show that the best strategy derived from our framework consistently outperforms baseline strategies when using 5-20% of the data. Moreover, training with just 10-20% of the data selected by our strategy yields superior results compared to training on the full dataset with the same time constraints. This demonstrates its practical efficiency for indoor novel view synthesis.

We summarize our contribution as follows:

1. We propose a framework for subset selection in novel view synthesis to address memory and time constraints encountered in real-life applications.

2. We propose a comprehensive measurement for the camera subset from the perspective of diversity, provide a theoretical analysis of the different utility functions, and validate their effectiveness through extensive empirical experiments.

3. We introduce INDOORTRAJ, a novel dataset designed for indoor novel view synthesis, featuring complex and extended trajectories that simulate intricate human behaviors.

## 2 Related work

### 2.1 Neural Rendering in Indoor Scenes

The development of neural rendering methods enables efficient dense reconstruction of 3D scenes and realistic novel view synthesis from 2D images (Lombardi et al., 2019; Mildenhall et al., 2020; Müller et al., 2022; Kerbl et al., 2023; Wang et al., 2021). Despite this progress, reconstructing indoor scenes poses distinct challenges. Indoor environments often contain significant occlusions between objects, textureless flat surfaces, and fine-grained small structures (Zhu et al., 2023). These characteristics complicate data collection, leading to long and intricate trajectories that usually contain a lot of redundant information. While some approaches focus on optimizing the model structures (Murez et al., 2020; Wang et al., 2023), another line of work addresses geometric difficulties by incorporating prior information, such as depth (Yu et al., 2022; Zhu et al., 2023; Xiang et al., 2024), normal prior (Yu et al., 2022; Zhu et al., 2023; Wang et al., 2022; Xiang et al., 2024), or semantic prior (Guo et al., 2022). In this work, we focus on efficiently utilizing cameras from a trajectory for neural rendering of indoor scenes. We design subset sampling strategies that enable the same base models (Kerbl et al., 2023; Müller et al., 2022) to achieve performance comparable to using the full frame set while utilizing only 10% to 20% of the data within a limited time constraint.

### 2.2 Sampling in Neural Rendering Methods

The reconstruction quality and convergence speed are strongly influenced by the sampling strategy. Sampling in 3D reconstruction can involve points, rays, or views. Earlier works such as Agarwal et al. (2011); Mendez et al. (2017) emphasize the need for an informative sampling strategy, particularly in large-scale and time-intensive traditional methods. Despite recent advancements in efficient neural rendering models, such as iNGP Müller et al. (2022) and 3DGS Kerbl et al. (2023), the need for efficient sampling persists, especially when the memory or time has some limit in real applications. A line of work focus on importance sampling on the points level along the rays, such as the representative NerfAcc solution sets (Li et al., 2023). Others explore sampling strategies at the ray level, as demonstrated in works such as Wu et al. (2023); Sun et al. (2024); Pais et al. (2025). However, relatively few works address the aspect of view subset sampling.

The most relevant line of research in the context of view subset sampling is the Next-Best-View (NBV) selection task, where heuristics are designed to optimize local information gain iteratively. Uncertainty-based heuristics (Pan et al., 2022; Goli et al., 2024; Lee et al., 2022) or warping consistency related heuristics (Ye et al., 2024) are particularly notable. Nonetheless, these approaches often require intensive computation, as they depend on a full or partial rendering procedure for each added camera. Additionally, they encounter challenges in scenarios where cameras are more freely distributed, as discussed in Kopanas & Drettakis (2023); Lee et al. (2023). An alternative branch of research employs diversity-related heuristics, focusing on factors such as spatial coverage (Kopanas & Drettakis, 2023) or Euclidean distances between camera pairs (Xiao et al., 2024; Pan et al., 2022). These strategies show promise for computational efficiency, but they are designed purely empirically, and the theoretical effectiveness of their objectives, combined with a greedy algorithm for identifying the best set of views, remains underexplored. In our work, we revisit this task as a subset selection problem. Specifically, we provide a comprehensive discussion of diversity-related measures in the context of view synthesis, exploring utility functions based on these measures and our task objective, leading to an efficient strategy for view subset sampling. Through both theoretical analysis and empirical studies, we demonstrate the effectiveness of our methods in novel view synthesis.

## 3 Preliminary

The following preliminary concepts will be used in the following sections. We will formally define and explain them below.

**Definition 1** (Submodularity). *A set function $z : 2^{\mathbb{D}} \to \mathbb{R}$ defined on a finite set $\mathbb{D}$ is called* submodular *if it satisfies the diminishing returns property:*

$$z(\mathbb{A} \cup \{k\}) - z(\mathbb{A}) \geq z(\mathbb{B} \cup \{k\}) - z(\mathbb{B})$$

---

**Algorithm 1** Greedy Algorithm for Maximizing Marginal Contribution

---

**Require:** Camera set $\mathbb{D}$, utility function $z$, subset size $K$
**Ensure:** A subset $\mathbb{S} \subseteq \mathbb{D}$ such that $|\mathbb{S}| = K$
 1: $\mathbb{S} \leftarrow \emptyset$                                   ▷ Initialize the solution set
 2: **for** $i = 1$ to $K$ **do**
 3:      $j^* \leftarrow \arg\max_{j \in \mathbb{D}\backslash\mathbb{S}} \; z(\mathbb{S} \cup \{j\}) - z(\mathbb{S})$
 4:      $\mathbb{S} \leftarrow \mathbb{S} \cup \{j^*\}$                             ▷ Add the selected element to $\mathbb{S}$
 5: **end for**
 6: **return** $\mathbb{S}$

---

Table 1: Distance measures that contribute to the diversity. Dist3D and Ang3D quantify spatial diversity, while the DistSem captures semantic diversity.

| Factor | Formula |
|---|---|
| Dist3D | $f_d(i,j) = \exp\left(-\frac{\|\boldsymbol{v}_i - \boldsymbol{v}_j\|^2}{2\sigma^2}\right)$, where $\boldsymbol{v}_i$ stands for the 3D position of camera $i$ |
| Ang3D | $f_a(i,j) = \frac{\text{trace}(\boldsymbol{R}_i^T \boldsymbol{R}_j) + 1}{4}$, where $\boldsymbol{R}_i$ stands for the rotation matrix of camera $i$ |
| DistSem | $f_p(i,j) = \frac{\boldsymbol{p}_i \cdot \boldsymbol{p}_j}{\|\boldsymbol{p}_i\|\|\boldsymbol{p}_j\|}$, where $\boldsymbol{p}_i$ stands for the image feature vector at camera $i$ |

*for all $\mathbb{A} \subseteq \mathbb{B} \subseteq \mathbb{D}$ and $k \in \mathbb{D} \setminus \mathbb{B}$.*

This diminishing returns property of functions, enabling greedy algorithms to achieve effective and provably near-optimal solutions (Nemhauser et al., 1978; Kulesza et al., 2012) for the NP-hard tasks defined by Problem 1.

**Definition 2** (Monotonicity). *A set function $z : 2^{\mathbb{D}} \to \mathbb{R}$ defined on a finite set $\mathbb{D}$ is called non-decreasing if for all $\mathbb{A} \subseteq \mathbb{B} \subseteq \mathbb{D}$, it holds that:*

$$z(\mathbb{A}) \leq z(\mathbb{B}).$$

This monotonicity (Feige et al., 2007; Nemhauser et al., 1978) is often discussed alongside submodularity, as the combination of these two properties enable the derivation of performance bounds for algorithms designed to solve Problem 1. Specifically, for monotone submodular functions, the greedy algorithm 1 provides an approximation guarantee by achieving an approximation ratio of $(e-1)/e$ (approximately 63.2% of the optimal value), as demonstrated by Nemhauser et al. (1978, Sec. 4). This algorithm is designed on marginal contribution of utility function $z(\mathbb{S})$.

### 3.1 Distance Measures

We propose a multifactor distance measure that efficiently integrates spatial, angular, and semantic distances, ensuring further diverse and informative view selection. We formulate these measures in Table 1. In particular, Dist3D and Ang3D are two fundamental measures in the spatial domain that target on position diversity and angular diversity respectively. Additionally, semantic distance offers prior information that helps emphasize the selection of visually diverse cameras, which can further enhance the global semantic information entropy. All measures are normalized within the range of 0 to 1, where a value of 1 indicates minimal distance and 0 indicates maximal distance.

Some works have considered the Euclidean distance between views as a factor that relevant to 3D reconstruction quality (Agarwal et al., 2011; Pan et al., 2022; Xiao et al., 2024), but rarely considers angular distance or semantic distance. We argue that neglecting this aspect can result in suboptimal outcomes. For example, when selecting two cameras out of three that provide similar observations and are positioned closely, as illustrated in the Ang3D scenario of Figure 1, the Ang3D measurement can guide the selection towards the camera pair with the greatest angular separation. Therefore, our approach integrates this measure with Dist3D and DistSem to establish a comprehensive distance measure. This promotes diverse sampling results

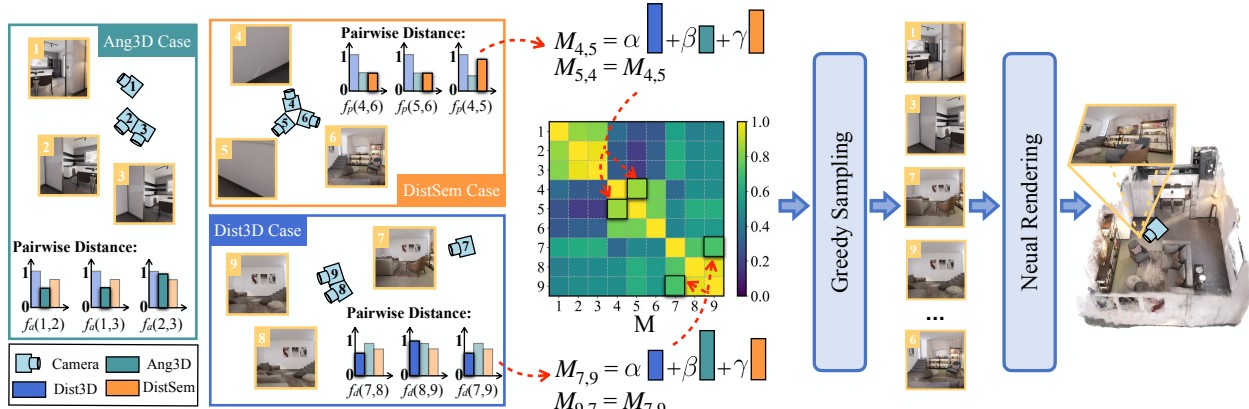

Figure 1: Data flow illustrating our subset selection framework. The depicted examples highlight how different factors in the distance measures influence the selection process when choosing two out of three cameras. The distance matrix is symmetric and integrates multiple factors, as defined in Table 1.

for the full view set $\mathbb{D}$ of size $N$. We define the multifactor distance matrix $\boldsymbol{M}_{\alpha,\beta} \in \mathbb{R}^{N \times N}$ as follows, with each component calculated in Table 1:

$$\boldsymbol{M}_{i,j} = \alpha \cdot f_d(i,j) + \beta \cdot f_a(i,j) + \gamma \cdot f_p(i,j), \quad i,j = 1, 2, \ldots, N, \tag{2}$$

where the corresponding weights satisfy $0 \leq \alpha, \beta, \gamma \leq 1$, $\alpha + \beta + \gamma = 1$. The impact of each factor on overall diversity can vary, depending on the specific weights assigned to them.

## 4 Utility Function Design for Subset Selection

In this section, we propose and analyze two utility functions as alternatives to $z(\mathbb{S})$ for the subset selection problem(Problem 1) based on our distance matrix $\boldsymbol{M}$. Additionally, we analyze a utility function derived from a representative baseline Kopanas & Drettakis (2023) with coverage measures.

### 4.1 Log Determinant Function

The first utility function is derived from Determinantal Point Process (DPP) (Kulesza et al., 2012), which is a probabilistic model designed for diverse subset sampling. The likelihood of selecting a subset is proportional to the determinant of the principal minor (sub-matrix) determined by the subset. Specifically, DPP defines a discrete probability distribution over the power set $2^{\mathbb{D}}$ through a positive semi-definite matrix $\boldsymbol{M} \in \mathbb{R}^{n \times n}$. For a subset $\mathbb{S} \subseteq \mathbb{D}$, the probability of selecting $\mathbb{S}$ is given by:

$$P(\mathbb{S}) = \frac{\det(\boldsymbol{M}_{\mathbb{S}})}{\det(\boldsymbol{M} + \boldsymbol{I})} \tag{3}$$

where $\boldsymbol{M}_{\mathbb{S}}$ denotes the submatrix of $\boldsymbol{M}$ with rows and columns selected according to the elements in $\mathbb{S}$ and $\boldsymbol{I}$ is the identity matrix. By definition, if $\boldsymbol{M}$ is a distance matrix, the repulsive property of the determinant encourages the DPP to prefer diverse subsets.

**Proposition 1.** *The log determinant function*

$$z_{\mathrm{DPP}}(\mathbb{S}) = \begin{cases} \log(\det P(\mathbb{S})) & \text{if } \mathbb{S} \neq \emptyset, \\ 1 & \text{if } \mathbb{S} = \emptyset. \end{cases} \tag{4}$$

*is submodular.*

The proof of submodularity is provided in Equation (74) (Kulesza et al., 2012).

## 4.2 Max-Min Distance Function

The second utility function is Max-Min Distance Function, we induct it from widely used heuristics in various domains. These include the farthest point sampling method in image acquisition (Eldar et al., 1997), Maximal Marginal Relevance(MMR) in information retrieval (Carbonell & Goldstein, 1998; Lin & Bilmes, 2011) and NBV methods (Pan et al., 2022; Xiao et al., 2024). And this can be adapted to distance measure between views for subset selection task. First, we need to define the Domain for the utility functions used afterward,

**Definition 3** (Domain of utility functions Defined by the Marginal Gain). *Let $\mathbb{X}$ be the domain of a utility function defined by the marginal gain. A nonempty set $\mathbb{S} \in \mathbb{X}$ if and only if there exists $k \in \mathbb{S}$ such that*

$$k = \underset{k^* \in \mathbb{D} \setminus (\mathbb{S} \setminus \{k^*\})}{\arg\max} \ z(\mathbb{S}) - z(\mathbb{S} \setminus \{k^*\}) \ and \ \mathbb{S} \setminus \{k\} \subseteq \mathbb{X}, \tag{5}$$

*where $z(\mathbb{S})$ is a utility function defined on set $\mathbb{S}$.*

Then, following the way of definition discussed by Lin & Bilmes (2011) and Ashkan et al. (2015), we can define the utility function as:

**Definition 4** (Max-Min Distance Function). *Define the set function $z_{\mathrm{DF}} : \mathbb{X} \to \mathbb{R}$, where the domain $\mathbb{X}$ follows the Definition 3 with $z_{\mathrm{DF}}$ as the utility function $z$. The function is recursively defined as:*

$$z_{\mathrm{DF}}(\mathbb{S} \cup \{k\}) - z_{\mathrm{DF}}(\mathbb{S}) = \min_{i \in \mathbb{S}}(1 - \boldsymbol{M}_{i,k}), \tag{6}$$

*and $z_{\mathrm{DF}}(\emptyset) = 0$. $\boldsymbol{M}_{i,k}$ is a distance measure between elements $i$ and $k$, $0 \le \boldsymbol{M}_{i,k} \le 1$.*

Using this definition, we establish the following proposition:

**Proposition 2.** *The function $z_{\mathrm{DF}}$ is monotone and submodular.*

*Proof.* To establish submodularity, we verify the diminishing returns property. Let $\mathbb{A} \subseteq \mathbb{B} \subseteq \mathbb{D}$ and consider an element $k \in \mathbb{D} \setminus \mathbb{B}$. Since $\mathbb{A} \subseteq \mathbb{B}$, it follows that any element in $\mathbb{A}$ is also in $\mathbb{B}$. Therefore, the set over which the minimum is taken for $\mathbb{B}$ contains all elements in the set over which the minimum is taken for $\mathbb{A}$. Such that the marginal contribution from adding $k$ to set $\mathbb{A}$ and $\mathbb{B}$ holds the following:

$$\min_{i \in \mathbb{A}}(1 - \boldsymbol{M}_{i,k}) \ge \min_{i \in \mathbb{B}}(1 - \boldsymbol{M}_{i,k}), \tag{7}$$

which consistent with the diminishing returns property.

To establish monotonicity, i.e. $z_{\mathrm{DF}}(\mathbb{A}) \le z_{\mathrm{DF}}(\mathbb{B})$, note that $0 \le \boldsymbol{M}_{i,k} \le 1$ by definition, which implies:

$$\min_{i \in \mathbb{B}}(1 - \boldsymbol{M}_{i,k}) \ge 0. \tag{8}$$

Since the marginal contribution is non-negative for any $k \in \mathbb{D} \setminus \mathbb{B}$, the function $z_{\mathrm{DF}}$ is monotone non-decreasing. $\square$

## 4.3 Uniform Coverage Function

Spatial coverage has been identified as a measure positively correlated with the quality of neural rendering, as highlighted by Kopanas & Drettakis (2023). To further explore its potential efficacy in the context of submodularity, this measure can be formalized into a utility function as follows:

**Definition 5** (Uniform Coverage Function). *Consider a finite set of disjoint regions in 3D space denoted by $\Omega$, where each view $i \in \mathbb{D}$ covers a subset of regions $A[i] \in \Omega$ determined by its frustum. Define the set function $z_{\mathrm{CF}} : \mathbb{X} \to \mathbb{R}$, where the domain $\mathbb{X}$ follows the Definition 3 with $z_{\mathrm{CF}}$ as the utility function $z$. The function is recursively defined as:*

$$z_{\mathrm{CF}}(\mathbb{S} \cup \{k\}) - z_{\mathrm{CF}}(\mathbb{S}) = \sum_{x \in \Omega} \left( \left( c_{\mathbb{S} \cup \{k\}}(x) \right)^\lambda + \left( 1 - TV_{\mathbb{S} \cup \{k\}}(x) \right) \right), \tag{9}$$

where $\lambda$ is the scaling factor and $z_{\text{CF}}(\emptyset) = 0$. $c_{\mathbb{S} \cup \{k\}}(x)$ denotes the ratio of views from $\mathbb{S} \cup \{k\}$ that cover region $x \in \Omega$, formally given by:

$$c_{\mathbb{S} \cup \{k\}}(x) = \frac{\sum_{i \in \mathbb{S} \cup \{k\}} \mathbb{1}_{obs}(i, x)}{|\mathbb{S} \cup \{k\}|}. \tag{10}$$

Function $\mathbb{1}_{obs}(i, x)$ is an indicator function that is 1 if view $i$ cover the region $x$, and 0 otherwise. The term $TV_{\mathbb{S} \cup \{k\}}(x)$ represents the total variation between the distribution of accumulated views over the angular space $\Theta$ at region $x$, given by:

$$TV_{\mathbb{S} \cup \{k\}}(x) = \frac{1}{2} \sum_{y \in \Theta} |p(x, y) - u(x)|, \tag{11}$$

where $p(x, y)$ denotes the probability density function (PDF) for direction $y$ at region $x$, and $u(x)$ represents the uniform PDF at $x$.

The function $z_{\text{CF}}$ is monotone since $z_{\text{CF}}(\mathbb{S} \cup \{k\}) - z_{\text{CF}}(\mathbb{S}) \geq 0$.

However, $z_{\text{CF}}$ is non-submodular. This can be observed by analyzing the two primary components of Equation 9. When the set size $\mathbb{S}$ increases, the value of Equation 10 can either increase or decrease, depending on whether the region $x$ is covered by the newly added views. Consequently, the value of the first component $\sum_{x \in \Omega} \left( c_{\mathbb{S} \cup \{k\}}(x) \right)^{\lambda}$ of Equation 9 depends on the balance between the portions of the increase and decrease. This violates the diminishing return in Definition 1. In addition, the $TV$ function in the second component $\sum_{x \in \Omega} \left( 1 - TV_{\mathbb{S} \cup \{k\}}(x) \right)$ of Equation 9 depends on how $k$ redistributes the views in the angular space. The marginal gain for larger set can exceed that for smaller set, if $k$ introduces uniformity to the larger set but increases non-uniformity for the smaller set.

This implies that the greedy algorithm on this non-submodular utility function may prioritize elements with high immediate gains but low overall contribution to the global optimum. This is particularly evident in the clustering phenomenon observed in the subset selection results, as demonstrated in Experiment 5.5.

## 5 Experiments

### 5.1 Dataset

To accurately assess the impact of our sampling strategy in indoor scenes, we select the widely-used **Replica dataset** (Straub et al., 2019) as one of our evaluation sets. However, in natural human motion captures, people may stop, move, or rapidly change direction. As a result, camera movements often do not maintain a consistent speed. These characteristics are not adequately captured by the Replica dataset. To complement this, we include two real-world datasets—**Mip-NeRF 360** (Barron et al., 2022) and **ScanNet** (Dai et al., 2017) — both captured by professional annotators and sharing some of Replica's characteristics, but still limited in their motion complexity. To address these limitations, we create a new dataset called **Indoor-Traj**, where the trajectories are captured within photorealistic indoor environments by real humans who have full control over the camera. The recorded trajectories feature long and complex camera movements. Additionally, a separate and disjoint testing trajectory is provided for each scene. This design, inspired by Warburg et al. (2023), ensures that the evaluation faithfully reflects the model's performance. More details can be found in the Appendix. A.

### 5.2 Baselines

In the experiments, we primarily compare three baseline sampling strategies with our proposed method. **The Random strategy** samples a random subset from the full dataset. **The Uniform strategy** samples frames at equal time intervals throughout the sequence. Starting from a random initial frame, this strategy samples frames at regular time intervals. If the end of the sequence is reached before selecting enough frames, it wraps around to continue sampling from the beginning.

The third baseline is about the greedy strategy with uniform coverage function, as introduced and analyzed in Section 4.3. The heuristic of uniform coverage was initially proposed in the NBV selection problem

Table 2: **Results on synthetic scenes:** quantitative evaluation on the testing scenes. We provide average rendering PSNRs and the standard deviations on five random seeds while sampling. The evaluation is performed on four scenes from the Replica dataset and an additional four scenes from our INDOORTRAJ dataset. The camera selection ratio is set to 5%.

| | Method | Replica Dataset | | | | IndoorTraj | | | |
|---|---|---|---|---|---|---|---|---|---|
| | | office-2 | office-3 | room-1 | room-2 | kitchen-1 | kitchen-2 | openplan-1 | living-1 |
| 3DGS | Random | 38.29± 0.59 | 39.65 ± 0.59 | 40.17 ± 0.66 | 42.23 ± 0.56 | 24.95 ± 0.33 | 23.5 ± 0.42 | 25.3 ± 0.22 | 24.4 ± 0.75 |
| | Uniform | 40.38 ± 0.14 | 41.09 ± 0.06 | 42.26 ± 0.12 | **43.33±0.07** | 25.45 ± 0.21 | 23.89 ± 0.2 | 26.12 ± 0.07 | 25.51 ± 0.2 |
| | $z_{\mathrm{CF}}$ | 35.33 ± 0.00 | 37.92 ± 0.00 | 38.63 ± 0.00 | 39.64 ± 0.00 | 26.96 ± 0.00 | 24.0 ± 0.00 | 26.55 ± 0.00 | 25.74 ± 0.00 |
| | $z_{\mathrm{DDP}}$ | 39.64 ± 0.38 | 40.84 ± 0.21 | 41.86 ± 0.31 | 42.84 ± 0.19 | 27.03 ± 0.18 | 24.63 ± 0.37 | 26.41 ± 0.09 | 26.69 ± 0.29 |
| | $z_{\mathrm{DF}}$ | **40.46±0.20** | **41.25±0.11** | **42.32±0.16** | 43.28 ± 0.22 | **27.44±0.09** | **25.58±0.27** | **26.78±0.05** | **27.09±0.32** |
| iNGP | Random | 35.79 ± 0.64 | 36.84 ± 0.15 | 36.77 ± 0.69 | 38.61 ± 0.34 | 24.83 ± 0.23 | 21.95 ± 0.63 | 24.75 ± 0.1 | 21.59 ± 0.61 |
| | Uniform | 37.38 ± 0.07 | 37.43 ± 0.04 | 38.25 ± 0.14 | **39.22±0.07** | 25.49 ± 0.15 | 21.76 ± 0.22 | 25.12 ± 0.09 | 22.82 ± 0.28 |
| | $z_{\mathrm{CF}}$ | 34.59 ± 0.00 | 36.12 ± 0.0 | 35.44 ± 0.00 | 37.67 ± 0.00 | 26.65 ± 0.00 | 23.22 ± 0.00 | 25.82 ± 0.00 | 22.61 ± 0.00 |
| | $z_{\mathrm{DDP}}$ | 36.73 ± 0.35 | 37.24 ± 0.12 | 37.75 ± 0.40 | 38.91 ± 0.25 | 26.45 ± 0.23 | 23.14 ± 0.41 | 25.44 ± 0.11 | 23.53 ± 0.30 |
| | $z_{\mathrm{DF}}$ | **37.45±0.08** | **37.47±0.07** | **38.29±0.25** | 39.13 ± 0.07 | **26.96±0.14** | **23.30±0.15** | **25.86±0.09** | **24.02±0.10** |

Table 3: **Results on real-world scenes:** quantitative evaluation on the Mip-NeRF 360 indoor scenes and ScanNet scenes with 3DGS. The camera selection ratio for Mip-NeRF 360 is set to 20% instead of our standard 5% setting, as the original views are already sparse.

| Method | Mip-NeRF 360 | | | | ScanNet | | | |
|---|---|---|---|---|---|---|---|---|
| | bonsai | counter | kitchen | room | scene0050_00 | scene0073_01 | scene0085_00 | scene0134_02 |
| Random | 26.91±0.60 | 23.66±0.46 | 25.45 ± 0.65 | 26.07±1.12 | 20.83 ± 0.48 | 23.10 ± 0.42 | 25.57 ± 0.25 | 21.45 ± 0.48 |
| Uniform | 28.94±0.15 | **24.82±0.25** | 26.36 ± 0.48 | 26.57±0.56 | **22.43 ± 0.25** | 24.59 ± 0.17 | **26.58 ± 0.11** | 22.00 ± 0.95 |
| $z_{\mathrm{CF}}$ | 28.07± 0.00 | 24.33± 0.00 | 26.45± 0.00 | 27.21± 0.00 | 20.58 ± 0.00 | 23.91 ± 0.00 | 6.27 ± 0.00 | 8.22 ± 0.00 |
| $z_{\mathrm{DDP}}$ | 27.71±0.59 | 24.28±0.30 | 25.86±0.48 | 26.56±0.45 | 21.25 ± 0.32 | 25.06 ± 0.29 | 26.05 ± 0.52 | 22.34 ± 0.87 |
| $z_{\mathrm{DF}}$ | **29.11±0.16** | 24.79±0.39 | **26.67±0.50** | **27.34±0.46** | 22.16 ± 0.32 | **25.82±0.27** | 26.52 ± 0.26 | **23.18±0.55** |

by Kopanas & Drettakis (2023). This proposed heuristic aligns well with our subset selection framework. It quantifies 3D diversity using a composite coverage measure, which introduces strong dependencies in sequential sampling. We select it as a representative for its line of work, and it is denoted by the utility function $z_{\mathrm{CF}}$ in the experiment section.

## 5.3 Implementation Details

The experiments are conducted using a single NVIDIA P100 GPU for 30,000 iterations. We utilize the default hyperparameters recommended by the two classical neural rendering methods, instant-NGP (iNGP) (Müller et al., 2022) and Gaussian Splatting (3DGS) (Kerbl et al., 2023). The hyperparameters $\alpha$ and $\beta$ of each factor in distance matrix $\boldsymbol{M}$ controlled the general relative importance among different measures, we set up a separate evaluation set with the four out of eight scenes in the Replica dataset (Straub et al., 2019) and empirically tune these two parameters. They are set to 0.7 and 0.2 respectively.

All the camera locations are scaled to the range [-4,4]. We set $\sigma = 0.5$ in the Gaussian kernel Dist3D to ensure smooth distance measurements. For DistSem measure, the image features are encoded using CLIP (Radford et al., 2021) trained on WebLI following Zhai et al. (2023). Since the axis aligned bounding box (AABB) scale significantly impacts the performance of iNGP (Müller et al., 2022), we set the value to 16 for the Replica dataset and 4 for others to achieve reasonable reconstruction results.

## 5.4 Main Results

In Table 2 and Table 3, we evaluate the rendering quality of three baseline sampling strategies as well as ours with the utility function $z_{\mathrm{DDP}}$ and $z_{\mathrm{DF}}$ on both synthetic scenes and real scenes. Except for the greedy sampling with the utility function $z_{\mathrm{CF}}$ (Kopanas & Drettakis, 2023), the subsets generated by the other four

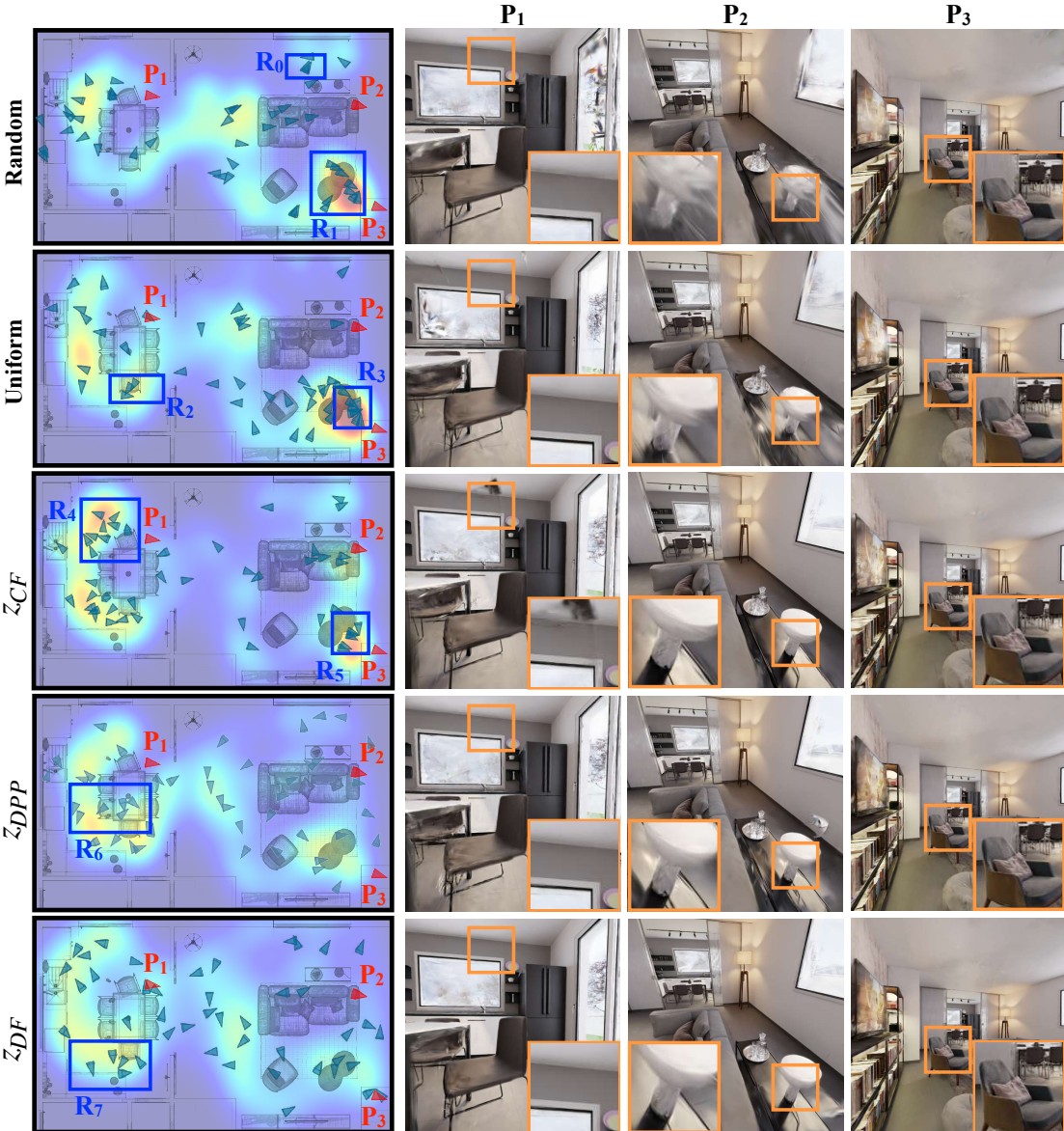

Figure 2: **Qualitative results on scene openplan-1** with a camera selection ratio of 5%: The first column presents a top-down heatmap at the same scale, showing the density of selected camera positions. For clearer demonstration, a 30% subset of the selected cameras from each method is drawn on the heatmap. The camera poses of the testing views marked in red are labeled as $P_1$, $P_2$, and $P_3$.

methods are influenced by random seed. To ensure robustness, we generate five sample sets for each method and report the average rendering Peak Signal-to-Noise Ratio (PSNR) along with the standard deviation.

For the standard Replica dataset, the camera movement is smooth with a largely uniform speed within a small region. We can find from Table 2 that the greedy strategy with $z_{DF}$ performs comparably to the Uniform strategy, offering a slight performance improvement on average. In contrast, the greedy method with $z_{CF}$ (Kopanas & Drettakis, 2023) underperforms significantly on the Replica dataset, showing an average of 2.25 PSNR drop even compared to the Random selection strategy. In real-world scenes, such as 5% samples on ScanNet 0085_00 and 0134_02 in Table 3, the $z_{CF}$ utility function (Kopanas & Drettakis, 2023) can sometimes lead to situations where there are not enough informative views for effective reconstruction.

On the more complex INDOORTRAJ dataset, the Uniform strategy is not effective anymore. The experiments show that, with the same sample size, the average PSNR gap between the Uniform strategy and our greedy solution with $z_{\mathrm{DF}}$ is approximately 1.46. Due to the more diverse and natural camera movement, the solution with $z_{\mathrm{CF}}$ (Kopanas & Drettakis, 2023) begins to outperform both the Uniform and Random strategies. However, it still results in an average 0.89 PSNR lower than the method with $z_{\mathrm{DF}}$. Specifically, in the scene *living-1*, this utility function $z_{\mathrm{CF}}$ even has 0.9 PSNR worse than $z_{\mathrm{DDP}}$ and 1.5 PSNR worse than $z_{\mathrm{DF}}$.

Notably, our greedy method using $z_{\mathrm{DDP}}$ consistently underperforms relative to $z_{\mathrm{DF}}$. This discrepancy can be attributed to two main measures. First, $z_{\mathrm{DDP}}$ lacks the monotonicity property, which limits its effectiveness during greedy sampling (Kulesza et al., 2012). Second, numerical precision issues in floating-point computations may contribute to the difficulty in distinguishing samples that are extremely close in the feature space, potentially leading to suboptimal selections.

Additionally, we observe a consistently large performance gap between the Random strategy and the Uniform strategy. Intuitively, the Random strategy can lead to clusters of cameras with similar poses capturing redundant scene information. A more theoretical explanation is related to Quasi-Monte Carlo (QMC) theory (Niederreiter, 1978), which demonstrates that the effectiveness of specific sampling patterns in a function approximation problem is theoretically superior to pure Monte Carlo sampling. We highlight this concept here as a reference for those interested in exploring this theory further.

## 5.5 Qualitative Evaluation

In addition to the quantitative evaluation, we also conduct an analysis of how the selection strategy behaves in the specific scenario. We select the scene openplan-1 as an example. This scene is the largest one within our INDOORTRAJ dataset, which includes a long training trajectory that spans two rooms.

**Camera distribution:** In the first column of Figure 2, we present a top-down heatmap showing the density of selected camera positions. **Our solution with $z_{\mathrm{DF}}$** shows a smoother coverage of the entire scene. Even in relatively dense regions like $R_7$, the selected cameras exhibit diverse orientations throughout the area. Instead, **greedy solution with $z_{\mathrm{CF}}$** (Kopanas & Drettakis, 2023) tends to densely select the cameras distributed on the corners of the scene, and there are multiple cameras having very similar poses (location and orientation) as seen in region $R_4$ and region $R_5$. This non-submodular metric prioritizes locally optimal solutions by favoring cameras whose frustum covers a larger portion of the scene's bounding box. **The non-monotone submodular metric** $z_{\mathrm{DDP}}$ exhibits a slight clustering effect in region $R_6$. The **Uniform strategy** largely relies on the moving speed of the camera. If the trajectory is captured with flexible speeds, as is common in user-generated data, the method will be biased toward the temporally dense area, such as region $R_2$ and region $R_3$. The **Random strategy** often samples cameras with similar poses, such as those in the region $R_1$, and even highly overlapping ones, as seen in $R_0$.

**Rendering Quality:** From the testing view at $P_1$ the observation aligns with what we identified with $z_{\mathrm{CF}}$. This metric tends to select cameras positioned at corners facing large open spaces, often overlooking scene boundaries. The slight clustering effect caused by the metric $z_{\mathrm{DDP}}$ results in poor quality in missing regions, such as the bottom of the chair in the $P_1$ view. For the Random and Uniform strategy, on the other hand, sampling is based directly on temporal factors. Consequently, both methods show a sparse distribution in the faster-moving parts of the scene. This sparse distribution causes a lack of cameras to render high-quality objects in these areas, such as the lamp in the testing view $P_2$. In the region around the testing view $P_3$, the Random, Uniform, and $z_{\mathrm{CF}}$ strategies concentrate more densely than our approach with $z_{\mathrm{DDP}}$ and $z_{\mathrm{DF}}$. However, despite the lower camera density, our strategies still achieve equally good rendering quality in this region. This outcome suggests a certain level of redundancy in the other three methods.

## 5.6 Performance Gaps with Varying Sample Ratios

In this experiment, we assess the rendering performance of various sampling strategies across different sample ratios. Specifically, we evaluate sample ratios of [0.01, 0.03, 0.05, 0.10, 0.15, 0.20]. For reference, the performance of the model trained on the full dataset over 30,000 iterations is indicated by a dashed line in Figure 3. The base neural rendering model used in these experiments is 3DGS (Kerbl et al., 2023).

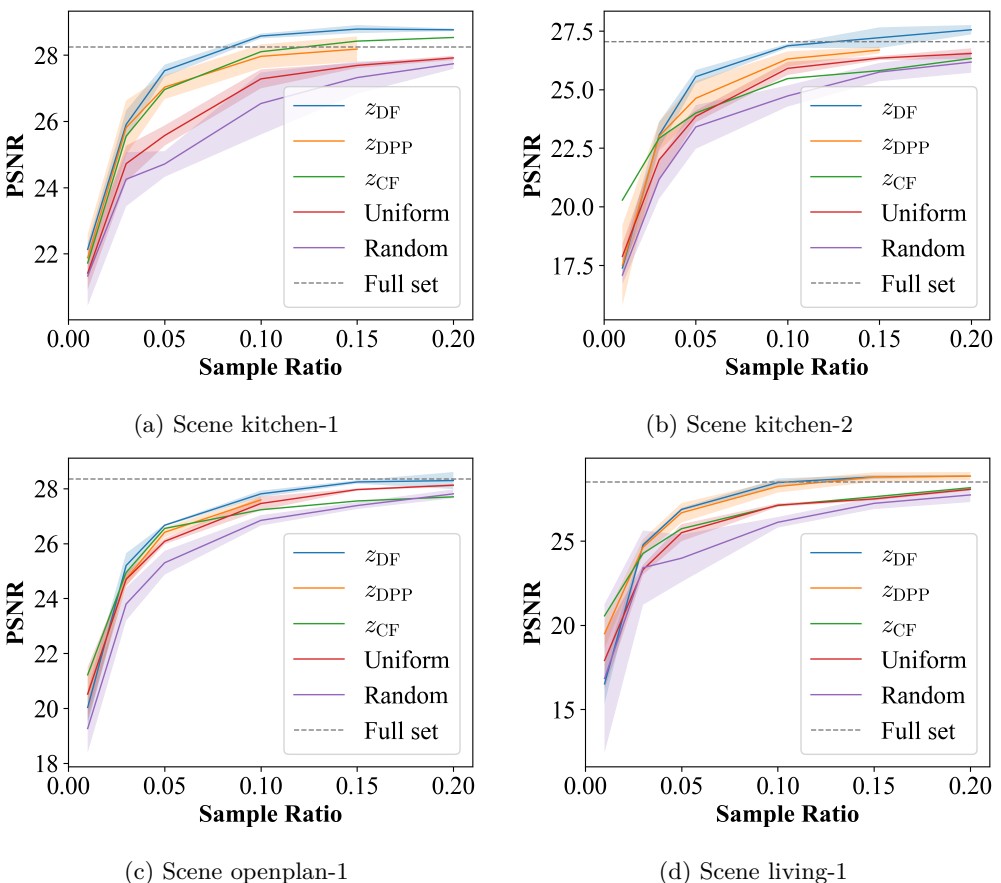

(a) Scene kitchen-1           (b) Scene kitchen-2

(c) Scene openplan-1          (d) Scene living-1

Figure 3: **Performance across different sample ratios.** The experiments are set on IndoorTraj dataset. Results are based on 30k iterations of training using the Gaussian Splatting backbone across different sampling strategies.

As shown in Figure 3, the approach with the utility function $z_{DF}$ and $z_{DPP}$ outperforms other strategies in three out of four scenes when sample ratio is larger than 0.03. The greedy solution with $z_{DF}$ achieves on average 0.8 higher PSNR for kitchen-1 and kitchen-2 scenarios using just 20% of data samples. Impressively, it matches or even exceeds the performance of models trained with the full dataset with the same 30k iterations. Notably, as shown in Figure 3a, it surpasses the full dataset model by 0.2 PSNR using only 10% of the data, and this advantage grows to 0.6 PSNR at 20%. Similar trends are observed in Figure 3b and 3d.

The greedy approach with baseline utility function $z_{CF}$ outperforms other strategies when the sample ratio is below 0.03. This result suggests that the frustum coverage measure is effective for extremely small subsets. However, its performance deteriorates as the sample ratio increases in most scenes. Additionally, this strategy becomes computationally expensive, with sampling time increasing exponentially with respect to the sample ratio. It requires over 100 times more sampling time compared to strategies using $z_{DF}$ and $z_{DPP}$.

We can also observe that the solution with $z_{DPP}$ is significantly constrained by the numerical precision issues. The metric becomes unsolvable even with 64-bit floating-point precision for an instance sample ratio of 0.2 in the cases of kitchen-1, kitchen-2, and openplan-1.

The underperformance of the model trained on the full dataset with a 30k iteration training budget suggests underfitting, which points to the presence of redundancy. We can also observe that the training trajectories across different scenes differ in the levels of redundancy, explaining why the performances converge at different sample ratios. These results highlight the necessity of crafting effective sampling strategies, especially for

efficient novel view synthesis involving extensively long trajectories. Our method can serve as a solid baseline for such strategies.

### 5.7 Abalation Study

To better understand the contribution of each component in our multifactor distance matrix, we performed an ablation study. By comparing the results, we can quantify the importance of each aspect in the overall system for current evaluation data. The experiment setting follows the main experiment, and we report the average PSNR in Table 4. We evaluate our best-performed solution with metric $z_{\mathrm{DF}}$. Each row in the table represents a different combination of measures included in the distance matrix.

From Table 4, we can observe that the model with all three measures consistently performs best across most scenes, indicating the importance of all three measures working together. For each individual measure, Dist3D proves to be the most informative across all scenes. For specific scenes, such as office-3, kitchen-1, and living-1, a distance matrix that includes only Dist3D achieves performance comparable to the multifactor version. We can observe that, the combination of AngDist and Dist3D typically yields the second/third-best rendering results, with a performance gap of less than 0.2 PSNR on average compared to the full distance matrix. Therefore, in real-world applications, prioritizing time efficiency, most cases can be effectively handled by combining AngDist and Dist3D.

Table 4: **Ablation study on distance measures.** We experiment various combinations of measures and compare the PSNRs of the rendering results.

| Method | Measures | | | Replica Dataset | | | | IndoorTraj | | | |
|---|---|---|---|---|---|---|---|---|---|---|---|
| | DistSem | AngDist | Dist3D | office-2 | office-3 | room-1 | room-2 | kitchen-1 | kitchen-2 | openplan-1 | living-1 |
| 3DGS | ✓ | ✗ | ✗ | 38.93 | 40.74 | 40.91 | 42.24 | 22.57 | 23.4 | 25.33 | 23.83 |
| | ✗ | ✓ | ✗ | 38.26 | 40.20 | 41.45 | 42.89 | 24.68 | 22.14 | 25.77 | 23.12 |
| | ✗ | ✗ | ✓ | 40.3 | 41.21 | 42.02 | 43.17 | 27.41 | 25.31 | 26.45 | **27.16** |
| | ✗ | ✓ | ✓ | 40.41 | 41.19 | 42.26 | 43.24 | 27.38 | 25.26 | 26.77 | 26.94 |
| | ✓ | ✓ | ✓ | **40.46** | **41.25** | **42.32** | **43.28** | **27.44** | **25.58** | **26.78** | 27.09 |
| iNGP | ✓ | ✗ | ✗ | 35.82 | 37.3 | 37.28 | 38.58 | 24.02 | 22.2 | 24.86 | 22.14 |
| | ✗ | ✓ | ✗ | 35.61 | 36.89 | 37.66 | 38.9 | 24.93 | 20.75 | 24.87 | 21.59 |
| | ✗ | ✗ | ✓ | 37.22 | 37.43 | 37.9 | 39.09 | 26.84 | 23.01 | 25.69 | 23.84 |
| | ✗ | ✓ | ✓ | 37.41 | 37.45 | 38.22 | 39.11 | 26.89 | **23.32** | 25.81 | 23.97 |
| | ✓ | ✓ | ✓ | **37.45** | **37.47** | **38.29** | **39.13** | **26.96** | 23.30 | **25.86** | **24.02** |

## 6 Conclusion

This study introduces a novel view subset selection framework for indoor novel view synthesis from monocular frames. The framework leverages several well-defined metrics designed from diversity-based measurement, enabling a more informed and efficient selection of views. To enable more comprehensive experiments that better simulate complex human capture behaviors, we introduce a new dataset called INDOORTRAJ. Through theoretical analysis and extensive experimentation, we demonstrate the effectiveness and practicality of the proposed strategies, highlighting their ability to enhance efficiency and scalability in indoor scene modeling.

## 7 Acknowledgement

This project is supported by Flanders AI Research Program and Methusalem Lifelines. HZ is supported by the China Scholarship Council.

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

# A   IndoorTraj dataset

We collect data from four realistic indoor scenes in Blender, each featuring objects with varying surface roughness and natural light reflection effects. Each scene includes a long training trajectory that spans the entire space. Annotators are tasked with controlling a 6-DOF (Degrees of Freedom) camera to navigate through virtual environments. Camera translation is managed using the WASD keys on the keyboard, while rotation is controlled through mouse movement. The trajectories are recorded at 24 frames per second with 1024x1024 resolution. To ensure diversity in camera movements and capture preferences, the trajectories for the training and testing splits are recorded by different annotators. The dataset statistics are as follows:

|       | kitchen-1 | kitchen-2 | openplan-1 | living-1 |
|-------|-----------|-----------|------------|----------|
| Train | 2253      | 3084      | 3212       | 2632     |
| Test  | 133       | 99        | 182        | 102      |

Table 5: Data size for INDOORTRAJ dataset. We render 10 frames per second for training set and 1 frame per second for testing set.

These scenes also feature diverse objects, such as semi-transparent glasses and highly reflective surfaces, as shown in the example frames in Figure 4. These characteristics present additional challenges for accurate novel view synthesis. In Figure 5, we provide the top-down trajectories of each scene and example rendering views.

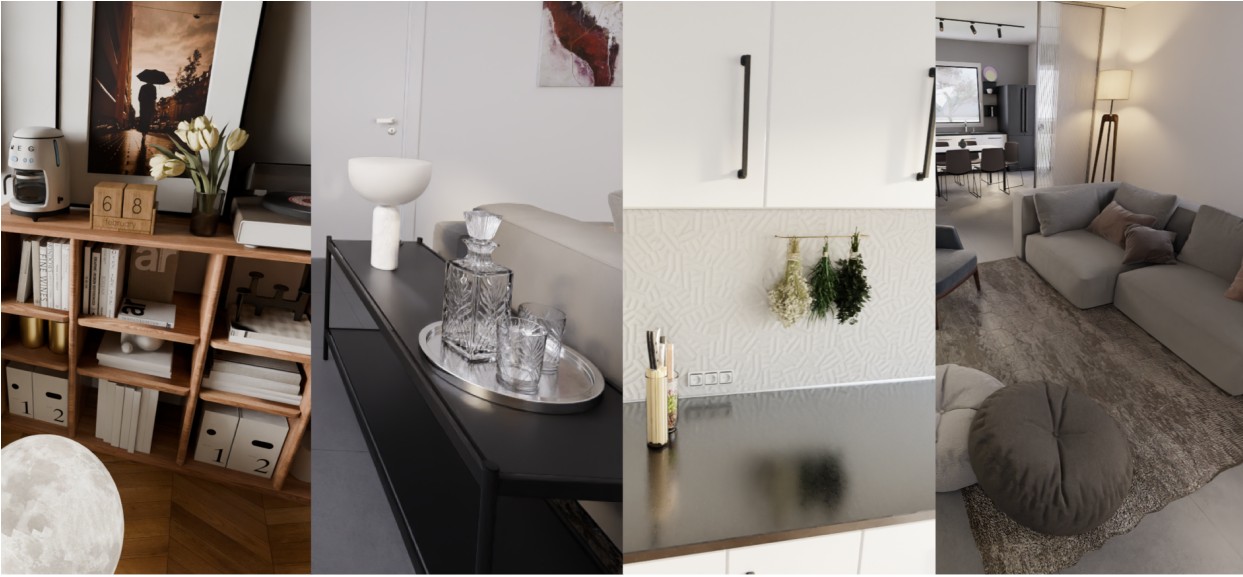

Figure 4: Visual characteristics in the INDOORTRAJ dataset, featuring object arrangement complexity, transparent objects, highly reflective surfaces, and areas with intricate textures.

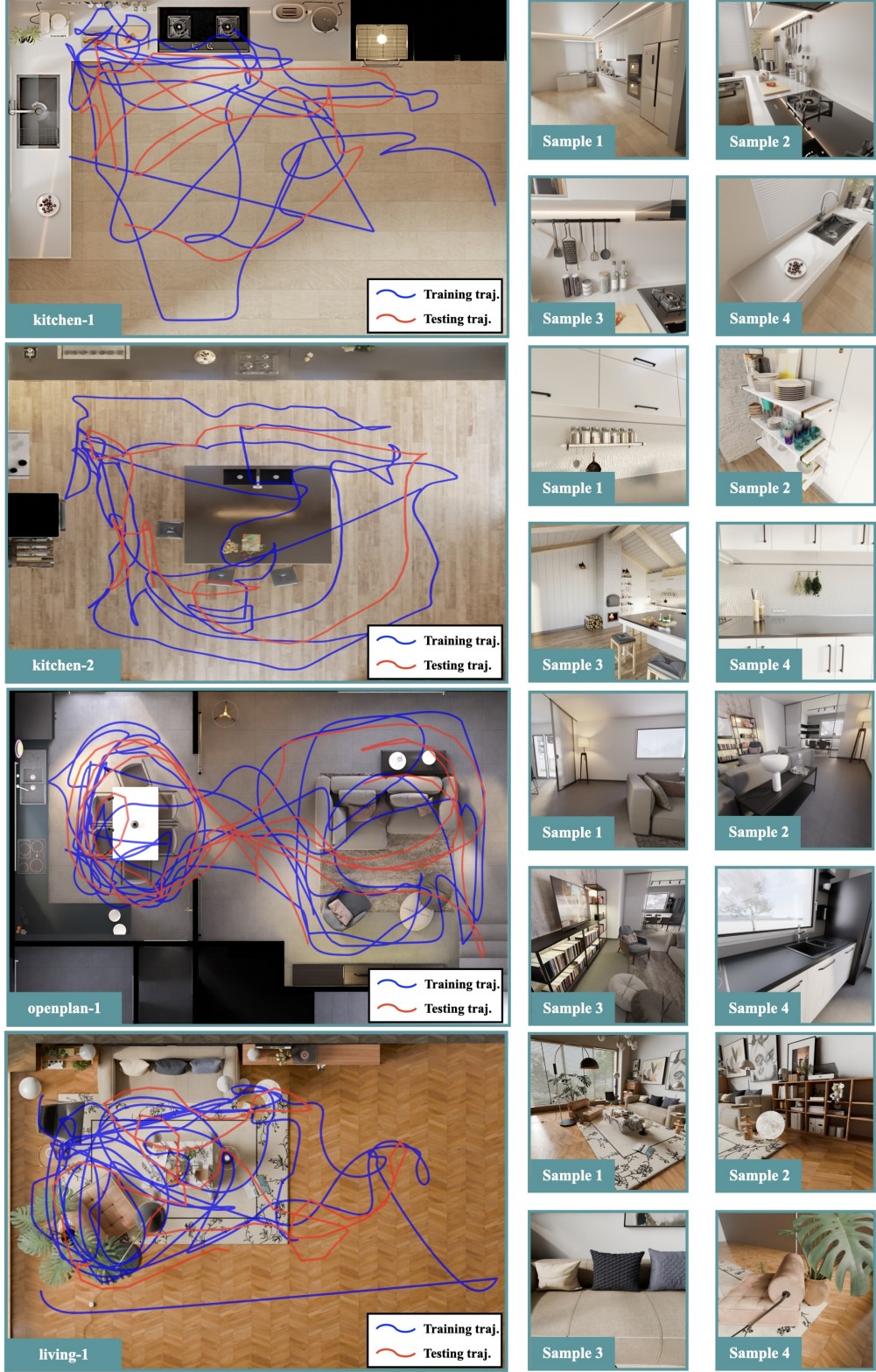

Figure 5: INDOORTRAJ dataset: Training and testing trajectories and example views

## B    Analysis on Hyperparameters

We selected the hyperparameters through grid search on four tuning scenes in Replica, as mentioned in implementation details. We found that this set of parameters consistently achieves good performance across different datasets in the main results. For the discussion on parameter sensitivity, we provide grid search tables of two representative scenes, *i.e.* Replica-Office4 and Replica-Office0 in Figure 6.

Although different scenes under the same sample ratio perform different range of PSNRs, we still observe that the optimal parameters are generally $\alpha = 0.7$, $\beta = 0.2$, and $\gamma = 0.1$. The results showed relatively low sensitivity to variations around the chosen parameters. Therefore, we empirically chose this set of parameters.

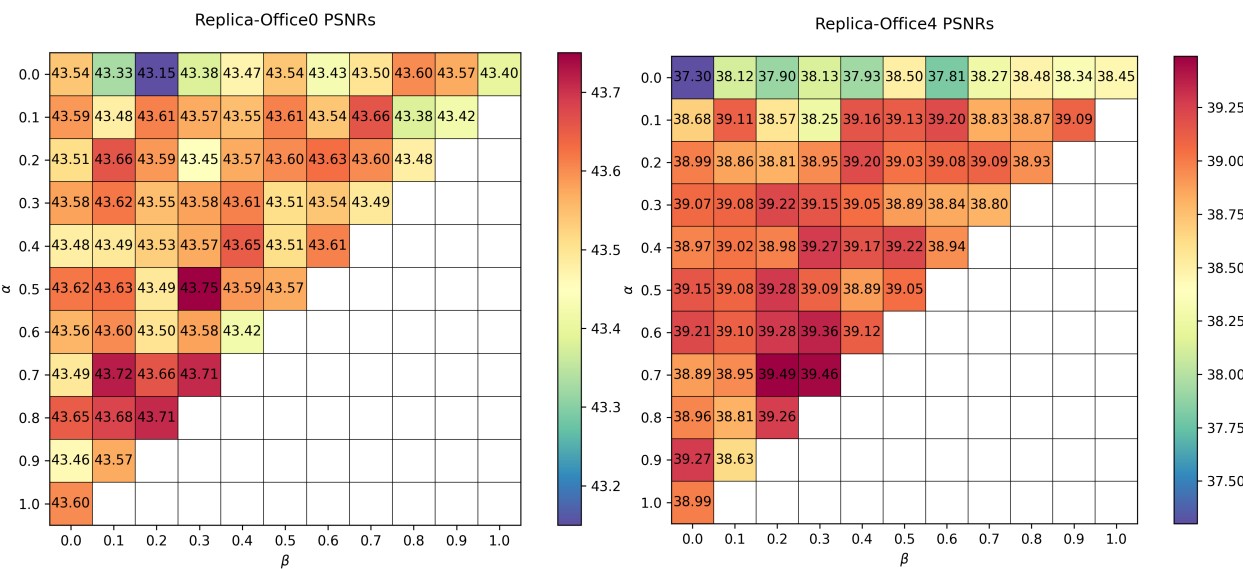

Figure 6: PSNRs grid search table for hyperparameters on tuning scenes Replica Office0 and Office 4.

## C    Time efficiency

Regarding the subset sampling time, we provide a reference table below. The sampling time for the method $z_{\text{CF}}$ proposed in Kopanas & Drettakis (2023) is significantly higher. Applying this method in real-world scenarios may require additional effort to optimize its efficiency at the code implementation stage. For $z_{\text{DPP}}$ and $z_{\text{DF}}$, the sampling times are slightly slower than Uniform and Random sampling, but this introduces only minor delay to the overall training time.

|                  | Replica | IndoorTraj | ScanNet | MipNeRF360 |
|------------------|---------|------------|---------|------------|
| Random           | <1s     | <1s        | <1s     | <1s        |
| Uniform          | <1s     | <1s        | <1s     | <1s        |
| $z_{\text{CF}}$  | 239s    | 325s       | 184s    | 96s        |
| $z_{\text{DPP}}$ | 2.32s   | 6.18s      | 1.02s   | <1s        |
| $z_{\text{DF}}$  | 1.73s   | 2.78s      | 1.48s   | <1s        |

