# OpenReview forum: "Diversity-Driven View Subset Selection for Indoor Novel View Synthesis"
_TMLR — Accepted by TMLR_

### Review · Reviewer_7FME · 2025-02-17

**Summary Of Contributions:**

The authors propose a framework for subset selection in indoor scene reconstruction task. To address memory and time constraints, they propose three metric functions to select only a subset of images for model training. Theoretical analysis of the metric functions is provided and experiments on several indoor scenes validate the effectiveness of the proposed method.

**Audience:**

Yes

**Claims And Evidence:**

No

**Requested Changes:**

Please refer to weaknesses.

**Strengths And Weaknesses:**

Strengths:
1. The topic is interesting and practical.
2. The authors compare three difference metric functions for subset selection, and collect a dataset for testing their method.

Weakness:
1. The authors claim their subset selection method could reduce memory and time constrains in real-world application, but they didn't provide comparisons. For example, memory/optimization time of methods in Table 2 and Figure 3.
2. They only conduct experiments on 8 scenes, which is limited. Experiments on more scenes and other datasets should be provided. For example, RealEstate10k
3. Optimization time is related to optimization iterations, the authors should provide figures of PSNR v.s. iterations of training on their selected set and full set.

---

> ### Author Response · Authors · 2025-03-11
>
> **W1:** In Figure 3, we can compare the memory savings achieved by different methods when reaching the same PSNR within the same training time (30k iterations), presented in terms of subset size. For instance in the scenes of IndoorTraj, the percentage of memory required by $z_{\textnormal{DF}}$ comparing to other strategies are as follows
>
>
> |      | Random | Uniform | $z_{\textnormal{CF}}$ | $z_{\textnormal{DPP}}$|
> | -------- | -------- | -------- |  -------- | -------- |
> | kitchen-1     |    0.17  |    0.25  | 0.46| 0.54|
> | kitchen-2     |    0.33  |    0.42 | 0.37 | 0.62|
> | openplan-1     |   0.48   |   0.71   |0.44 | 0.90|
> | living-1     |     0.36 |    0.45  | 0.47| 0.99|
>
>
>
> We can also analyze the training curve to compare the time cost of method $z_{\textnormal{DF}}$ relative to other strategies when achieving the final PSNR under the same sample size 0.05. For instance, the following table illustrates this:
> |      | Random | Uniform | $z_{\textnormal{CF}}$ | $z_{\textnormal{DPP}}$|
> | -------- | -------- | -------- |  -------- | -------- |
> | kitchen-1     |  0.1 |   0.15   | 0.41 | 0.46|
> | kitchen-2     |    0.19 |   0.28 | 0.43| 0.48 |
> | openplan-1     |   0.23   |   0.48  | 0.87| 0.51|
> | living-1     |0.11 |   0.25   |  0.47| 0.61 |
>
> These results shows that sampling with metric function $z_{\textnormal{DF}}$ requires less time or less memory compared to other methods.
>
> ---
>
> Regarding the subset sampling time, we provide a reference table below. The sampling time for the method $z_{\text{CF}}$ proposed in [2] is significantly higher. Applying this method in real-world scenarios may require additional effort to optimize its efficiency at the code implementation stage. For $z_{\text{DPP}}$ and $z_{\text{DF}}$, the sampling times are slightly slower than Uniform and Random sampling, but this introduces only minor delay to the overall training time.
>
> |                     | Replica | IndoorTraj | ScanNet | MipNeRF360 |
> | --------            | -------- | -------- | -------- | -------- |
> | Random              | <1s     | <1s     | <1s| <1s|
> | Uniform             | <1s     | <1s     | <1s| <1s|
> | $z_{\text{CF}}$     |    239s  |    325s  | 184s| 96s |
> | $z_{\text{DPP}}$    |  2.32s    |   6.18s   |   1.02s  | <1s |
> | $z_{\text{DF}}$     | 1.73s     | 2.78s     | 1.48s | <1s|
>
>
> **W2:** For scenes in RealEstate10k, they are featured for short trajectory, high redundant views but sparse multi-view correspondence which requires advanced feature matching techniques [1] other than pure neural rendering baselines. But we expanded experiments to real-world scenes in ScanNet and MipNeRF360 instead as described in our general response.
>
> **W3:** We can provide the training curves of main results in the revision for readers reference. We provide an example of scene kitchen-1 in IndoorTraj evaluated every 4k iterations in the following table. :
>
> |      | 4k | 8k | 12k | 16k| ... | 30k|
> | -------- | -------- | -------- |  -------- | -------- | -------- | -------- |
> | Random                 |   24.00   |    24.58  | 24.84| 24.89| ...| 24.95|
> | Uniform                |   24.31   | 24.87   | 25.11 | 25.41| ...| 25.45|
> | $z_{\textnormal{CF}}$  | 24.88   |  26.00   | 26.41 | 26.53| ...| 26.69 |
> | $z_{\textnormal{DPP}}$ |  25.10    | 25.93  | 26.36| 26.92| ... | 27.03 |
> | $z_{\textnormal{DF}}$  | 25.44   |  26.52  | 26.99| 27.39|...| 27.44|
>
> **Reference:**
>
> [1] Chen, Y., Xu, H., Zheng, C., Zhuang, B., Pollefeys, M., Geiger, A., ... & Cai, J. (2024, September). Mvsplat: Efficient 3d gaussian splatting from sparse multi-view images. In *European Conference on Computer Vision* (pp. 370-386). Cham: Springer Nature Switzerland.
>
> [2] Kopanas, G., & Drettakis, G. (2023). Improving NeRF Quality by Progressive Camera Placement for Free-Viewpoint Navigation.

---

### Review · Reviewer_vC6f · 2025-02-24

**Summary Of Contributions:**

**-----Context-----**

This paper addresses several key challenges in view subset selection for novel view synthesis. Computing traditional rendering metrics like PSNR requires first reconstructing the scene from input images, making these metrics computationally expensive for optimal view subset selection where any possible subset of views could be considered. Additionally, selecting an optimal subset of K views that maximizes performance is an NP-hard problem. While some existing methods rely on time-consuming training iterations for view selection, and others focus primarily on view diversity through distance and frustum overlap, they fail to adequately capture angle variation and scene semantics.

**-----Contributions-----**

To address these limitations, the authors propose a comprehensive framework for efficient, greedy view subset selection that optimizes both memory usage and computational time.

Their approach introduces a novel multifactor distance measure that simultaneously promotes (1) spatial coverage, (2) directional diversity, and (3) content variation.

Authors investigate the mathematical properties of the proposed distance measure in order to provide theoretical guarantees on the performance of the greedy view subset selection algorithm.

Moreover, the authors contribute a new dataset of 4 scenes that better reflects complex human capture behaviors by incorporating greater variety in camera motion speed and trajectory length.

Authors show that their greedy view selection algorithm outperforms several baseline methods on their two datasets (including their own), and that the proposed measure appears to be a good proxy for traditional, expensive-to-compute rendering metrics.

**Audience:**

Yes

**Broader Impact Concerns:**

I did not identify any broader impact concerns.

**Claims And Evidence:**

Yes

**Requested Changes:**

**-----Critical for acceptance-----**

- Definition 3: The paper must absolutely clarify that the function $z_{DF}$ is well-defined on the collection of all subsets of $\\mathbb{D}$.

**-----Non Critical for acceptance-----**

- It would be useful to clearly explain mathematically what $\\mathbb{D}$ is at the beginning of the paper. I understand that $\\mathbb{D}$ is the set of all images $I_i \\in [0, 1]^{H_i \\times W_i \\times 3}$ but right now, $\\mathbb{D}$ is introduced without proper definition.

- Figure 1: *Neual Rendering* > *Neural Rendering*.

- Section 4.1: The sentence "*the determinant of the principal minor (sub-matrix)*" is not correct. A minor is, by definition, the determinant of a sub-matrix. The sentence should be "*the principal minor (determinant of sub-matrix)*".

- Proposition 1: Writing "log(det(P(S)))" is not correct as P(S) is a scalar probability, it should be "log(P(S))".

- Definition 3: What are the initial values, for $S = \\emptyset$ and $S = \\{k\\}$?

- Section 5.7: *Abalation Study* > *Ablation Study*

**Strengths And Weaknesses:**

**-----Strengths-----**

1. The paper is well-structured and addresses a significant yet often overlooked problem in the literature: selecting an optimal subset of views for novel view synthesis.

2. The proposed multifactor measure is interesting and effectively combines spatial coverage, directional diversity, and content variation. The quantitative results demonstrate that this approach outperforms all proposed baselines across multiple scenes.

3. I appreciate the paper's focus on providing theoretical analysis and guarantees for their approach. Such considerations are both interesting and important for the research community.

4. The comprehensive ablation study effectively demonstrates the importance and impact of each component in the proposed metric.

5. The introduction of IndoorTraj, a new dataset specifically designed for indoor novel view synthesis with complex camera trajectories, represents a valuable contribution to the research community.

**-----Weaknesses-----**
1. The quantitative evaluation is limited to only 8 scenes, which is insufficient for validating a view selection method. Given that the proposed method is applicable to any RGB image dataset, evaluation on very common larger-scale datasets (such as ScanNet) would strengthen the paper's claims, particularly for indoor environments where such datasets are readily available.

2. The qualitative evaluation lacks depth, with only a single example provided to demonstrate the performance of the method.

3. Several mathematical claims require additional justification or contain errors. Most notably, the set function introduced in Definition 3 is not properly defined. The function is described through an iterative process, but for it to be well-defined on all subsets of $\\mathbb{D}$, the value obtained with this iterative process should be order-independent: For a given subset $S=\\{I_1,...,I_k\\}$, the value of $z_{DF}(S)$ should not depend on the order of the elements $I_i$ used during computation. The paper neither proves this property nor addresses potential order dependency. It is actually possible to find counter-examples for which the function is not order-independent.

For instance, let's consider a set $\\mathbb{D} = \\{I_1, I_2, I_3\\}$ for which the distance matrix $M$ is defined as follows:
\\begin{equation}
    M = \\begin{bmatrix}
        0 & 0.5 & 0.4 \\\\
        0.5 & 0 & 0.9 \\\\
        0.4 & 0.9 & 0
    \\end{bmatrix}
\\end{equation}
The matrix $M$ is a proper distance measure with elements in $[0, 1]$, as required by Definition 3. Such matrix can trivially be obtained by taking $D \\subset \\mathbb{R}$ equipped with the euclidean distance, and $I_0 = 0$, $I_1 = -0.5$, and $I_2 = 0.4$.
The matrix $1-M$ is then equal to:
\\begin{equation}
    1-M = \\begin{bmatrix}
        1 & 0.5 & 0.6 \\\\
        0.5 & 1 & 0.1 \\\\
        0.6 & 0.1 & 1
    \\end{bmatrix}
\\end{equation}

Let's start with the subset $S = \\{0\\}$. The paper does not specify what the initial value of $S$ should be for a single element set, so let's denote $z_0 := z_{DF}(\\{0\\})$.

We can prove that depending on the order of the elements used for computing $z_{DF}(D)$, the result of the iterative process is different.

**Let's first consider the order $0, 1, 2$.**

We apply equation (5) with $S = \\{0\\}$, and $k = 1$.

We obtain $z_{DF}(\\{0, 1\\}) = z_{DF}(\\{0\\}) + \\min_{i \\in \\{0\\}} (1-M_{i,1}) = z_0 + 0.5$.

We then reapply equation (5) with $S = \\{0, 1\\}$, and $k = 2$.

We obtain $z_{DF}(\\{0, 1, 2\\}) = z_{DF}(\\{0, 1\\}) + \\min_{i \\in \\{0, 1\\}} (1-M_{i,2}) = z_{DF}(\\{0, 1\\}) + 0.1 = z_0 + 0.6$.

**Let's now consider the order $0, 2, 1$.**

We apply equation (5) with $S = \\{0\\}$, and $k = 2$.

We obtain $z_{DF}(\\{0, 2\\}) = z_0 + \\min_{i \\in \\{0\\}} (1-M_{i,2}) = z_0 + 0.6$.

We then reapply equation (5) with $S = \\{0, 2\\}$, and $k = 1$.

We obtain $z_{DF}(\\{0, 1, 2\\}) = z_{DF}(\\{0, 2\\}) + \\min_{i \\in \\{0, 2\\}} (1-M_{i,1}) = z_{DF}(\\{0, 2\\}) + 0.1 = z_0 + 0.7$.

It appears that the function $z_{DF}$ is not well-defined on the collection of all subsets of $\\mathbb{D}$. I acknowledge that I might have misunderstood the definition of $z_{DF}$ or missed an important detail, so I would be grateful if the authors could clarify this point.

If I'm mistaken and the function is well-defined, I would be glad to modify my review as the paper would then provide very interesting theoretical results on the view selection problem.
However, if my interpretation is correct, then the theoretical results and guarantees of the paper are not valid, which would be particularly concerning as the function $z_{DF}$ reaches the highest performance on the datasets and can be considered as the most important contribution of the paper.

---

> ### Author Response · Authors · 2025-03-11
>
> Thank you for your very thoughtful and thorough feedback.  It is clear that you spent a great amount of time understanding every step of the paper.
>
> **W1:** We have included additional experiments involving nine real-world scenes and expanded the discussion in our general response.
>
> **W2:** The primary goal of the qualitative evaluation is to provide a more intuitive visualization of different scenerios. These kind of scenerios are observable in other scenes. We will enhance the explaination of the visualization.
>
> In addtion, we acknowledge that some conclusions may seem evident to those familiar with the mechanics of each sampling strategy. For instance, in Random and Uniform sampling, a clustering effect is expected when the camera moves at highly variable speeds, as these methods sample based on temporal factors. Similarly, the sampling pattern of the $z_{\text{CF}}$ strategy can be anticipated. Since this method does not account for object occupancy and its metric function favors maximizing camera frustum coverage within the bounded region, it naturally emphasizes views from the corners pointing toward the center.

---

> ### Author Response · Authors · 2025-03-11
>
> **W3:** We follow the way of definition such as [1,2] in the method called Maximal Marginal Relevance (MMR), which defines the submodular function using the marginal gain. We agree that the definition of the domain for this type of set function should be further clarified. The set function $z(\mathbb{S})$ requires a valid set as input, which means the domain is constrainted by the set generation procedure. We revised the definition in the following way:
>
> > **Definition 3 (Domain of Metric Functions Defined by the Marginal Gain).** Let $\mathbf{X}$ be the domain of a metric function defined by the marginal gain. A nonempty set $\mathbb{S}\in \mathbf{X}$  if and only if there exists $k \in \mathbb{S}$ such that:
> > $$ k = \arg\max_{k^* \in \mathbb{D} \setminus (\mathbb{S} \setminus \{k^*\})} z(\mathbb{S}) - z(\mathbb{S} \setminus \{k^*\}) \text{ and } \mathbb{S} \setminus \{k\} \in \mathbf{X}, $$
> > where $z(\mathbb{S})$ is a metric function defined on the set $\mathbb{S}$.
>
> Then, following the way of definition discussed by [1,2], we can define the metric function as:
>
> > **Definition 4 (Max-Min Distance Function).** Define the set function $z_{\textnormal{DF}}: \mathbf{X} \to \mathbb{R}$, where the domain $\mathbf{X}$ follows the Definition 3 with $z_{\textnormal{DF}}$ as the metric function $z$. The function is recursively defined as:
> > $$z_{\textnormal{DF}}(\mathbb{S} \cup \{k\}) - z_{\textnormal{DF}}(\mathbb{S}) = \min_{i\in \mathbb{S}} (1-\mathcal{M}_{i,k}),$$
>
> > and $z_{\textnormal{DF}}(\emptyset)=0$. $\mathcal{M}_{i,k}$ is a distance measure between elements $i$ and $k$,
>
> > $ 0 \leq \mathcal{M}_{i,k} \leq 1 $.
>
>
> The definition of Uniform Coverage Function $z_{\textnormal{DF}}$ is also revised to define on domain $\mathbf{X} \to \mathbb{R}$.
>
> Returning to your examples, based on the above definitions, generating the sets in the order 0, 2, 1 is not possible. This is because the set {0, 2} serves as an intermediate state for constructing the set {0, 1, 2} in the order of 0,2,1. However, {0, 2} is not a valid set within the domain of the function $z_{\textnormal{DF}}$.
>
> **Reference:**
>
> [1] Hui Lin and Jeff Bilmes. A class of submodular functions for document summarization. In Proceedings of the 49th annual meeting of the association for computational linguistics: human language technologies, pp. 510–520, 2011.
>
> [2] Azin Ashkan, Branislav Kveton, Shlomo Berkovsky, and Zheng Wen. Optimal greedy diversity for recommendation. In IJCAI, volume 15, pp. 1742–1748, 2015.

---

### Review · Reviewer_bC5i · 2025-02-28

**Summary Of Contributions:**

This paper introduces a framework to improve efficiency in indoor novel view synthesis from monocular video sequences. By formulating the problem as a combinatorial optimization task, the authors develop diversity-based metric functions that account for spatial, angular, and semantic variations. They also introduce IndoorTraj, a novel dataset with complex human-like camera trajectories, demonstrating that their method achieves superior synthesis quality using only 5-20% of the data. Theoretical analysis and experiments validate that their greedy optimization strategy effectively balances computational efficiency and reconstruction performance.

**Audience:**

Yes

**Claims And Evidence:**

Yes

**Requested Changes:**

It would be good if authors can expand evaluation to additional datasets and analyzing sensitivity to hyperparameters in different environments.

**Strengths And Weaknesses:**

Strengths:

- The paper presents a well-structured combinatorial optimization approach to subset selection for novel view synthesis, addressing redundancy in monocular video sequences.

- The authors provide rigorous theoretical analysis of their metric functions and validate them with experiments.

-   The method significantly reduces data requirements while maintaining or improving rendering quality, making it practical for real-world applications.

- It introduces a new dataset (IndoorTraj) that features complex, realistic human-driven camera trajectories

- Experiments demonstrate consistent improvements over baseline strategies

Weaknesses:

- The framework is tested primarily on IndoorTraj and Replica, but its effectiveness on more diverse datasets or real-world capture scenarios is not deeply explored.

- Certain metric functions (e.g., log-determinant DPP) may still suffer from numerical precision issues or scalability challenges?

- The framework requires tuning parameters (such as $\alpha,\beta,\gamma$) in the distance matrix, and the robustness of these settings across different datasets is not fully analyzed.

---

> ### Author Response · Authors · 2025-03-11
>
> **W1**: We have included additional experiments involving nine real-world scenes and expanded the discussion in our general response.
>
> **W2**: Our paper does not aim to improve the stability or scalability of DPP methods; rather, we focus on applying this method to our specific setting. We applied the DPP algorithm following DPPy[3]. As noted in [1], [2] and [4], the DPP algorithm can suffer from numerical instabilities. Although adding a scaled identity matrix ($\epsilon I$) can mitigate numerical instability in DPP, it alters the value of the metric function, potentially steering the final results in an incorrect direction. We acknowledge that the exact DPP algorithm requires $O（n^3）$ time complexity, making it challenging to scale to large matrices. However, in our setting, we have not encountered scalability issues. For large-scale data, one can refer to more advanced solution as described in [4].
>
> **Reference:**
>
> [1] Bardenet, R., Lavancier, F., Mary, X., & Vasseur, A. (2017). On a few statistical applications of determinantal point processes. ESAIM: Proceedings and Surveys, 60, 180-202.
>
> [2] https://github.com/guilgautier/DPPy/issues/58
>
> [3] Gautier, G., Polito, G., Bardenet, R., & Valko, M. (2019). DPPy: DPP sampling with Python. Journal of Machine Learning Research, 20(180), 1-7.
>
> [4] Chen, Laming, Guoxin Zhang, and Eric Zhou. "Fast greedy map inference for determinantal point process to improve recommendation diversity." Advances in Neural Information Processing Systems 31 (2018).

---

> ### Author Response · Authors · 2025-03-11
>
> **W3**: We selected this set of parameters through grid search some scenes as mentioned in the paper, and we also found that this set of parameters consistently achieves good performance across different datasets in the main results. For the discussion on parameter sensitivity, we provide grid search tables of two representative scenes.
>
> Our key observation is that the optimal parameters are generally $\alpha = 0.7$, $\beta = 0.2$, and $\gamma = 0.1$. The results showed relatively low sensitivity to variations around the choosen parameters. Therefore, we empirically chose this set of parameters.
>
> **PSNRs for Subsets Sampled Using Different Hyperparameters on Replica-Office4**
>
> | $\alpha / \beta$ | 0                | 0.1   | 0.2              | 0.3              | 0.4   | 0.5   | 0.6   | 0.7   | 0.8   | 0.9   | 1.0   |
> | ---------------- | ---------------- | ----- | ---------------- | ---------------- | ----- | ----- | ----- | ----- | ----- | ----- | ----- |
> | **0**            | 37.3             | 38.12 | 37.9             | 38.13            | 37.93 | 38.5  | 37.81 | 38.27 | 38.48 | 38.34 | 38.45 |
> | **0.1**          | 38.68            | 39.11 | 38.57            | 38.25            | 39.16 | 39.13 | 39.2  | 38.83 | 38.87 | 39.09 |       |
> | **0.2**          | 38.99            | 38.86 | 38.81            | 38.95            | 39.2  | 39.03 | 39.08 | 39.09 | 38.93 |       |       |
> | **0.3**          | 39.07            | 39.08 | 39.22            | 39.15            | 39.05 | 38.89 | 38.84 | 38.8  |       |       |       |
> | **0.4**          | 38.97            | 39.02 | 38.98            | **39.27** | 39.17 | 39.22 | 38.94 |       |       |       |       |
> | **0.5**          | 39.15            | 39.08 | 39.28            | 39.09            | 38.89 | 39.05 |       |       |       |       |       |
> | **0.6**          | 39.21            | 39.1  | **39.28** | **39.36** | 39.12 |       |       |       |       |       |       |
> | **0.7**          | 38.89            | 38.95 | **39.49** | **39.46** |       |       |       |       |       |       |       |
> | **0.8**          | 38.96            | 38.81 | 39.26            |                  |       |       |       |       |       |       |       |
> | **0.9**          | **39.27** | 38.63 |                  |                  |       |       |       |       |       |       |       |
> | **1.0**          | 38.99            |       |                  |                  |       |       |       |       |       |       |       |
>
> **PSNRs for Subsets Sampled Using Different Hyperparameters on Replica-Office0**
>
> | $\alpha / \beta$ | 0     | 0.1              | 0.2              | 0.3              | 0.4   | 0.5   | 0.6   | 0.7              | 0.8   | 0.9   | 1.0   |
> | ---------------- | ----- | ---------------- | ---------------- | ---------------- | ----- | ----- | ----- | ---------------- | ----- | ----- | ----- |
> | **0**            | 43.54 | 43.33            | 43.15            | 43.38            | 43.47 | 43.54 | 43.43 | 43.50            | 43.60 | 43.57 | 43.40 |
> | **0.1**          | 43.59 | 43.48            | 43.61            | 43.57            | 43.55 | 43.61 | 43.54 | **43.66** | 43.38 | 43.42 |       |
> | **0.2**          | 43.51 | **43.66** | 43.59            | 43.45            | 43.57 | 43.60 | 43.63 | 43.60            | 43.48 |       |       |
> | **0.3**          | 43.58 | 43.62            | 43.55            | 43.58            | 43.61 | 43.51 | 43.54 | 43.49            |       |       |       |
> | **0.4**          | 43.48 | 43.49            | 43.53            | 43.57            | 43.65 | 43.51 | 43.61 |                  |       |       |       |
> | **0.5**          | 43.62 | 43.63            | 43.49            | **43.75** | 43.59 | 43.57 |       |                  |       |       |       |
> | **0.6**          | 43.56 | 43.60            | 43.50            | 43.58            | 43.42 |       |       |                  |       |       |       |
> | **0.7**          | 43.49 | **43.72** | 43.66            | **43.71** |       |       |       |                  |       |       |       |
> | **0.8**          | 43.65 | 43.68            | **43.71** |                  |       |       |       |                  |       |       |       |
> | **0.9**          | 43.46 | 43.57            |                  |                  |       |       |       |                  |       |       |       |
> | **1.0**          | 43.60 |                  |                  |                  |       |       |       |                  |       |       |       |

---

### Author Response · Authors · 2025-03-11
**General Response: Experiments with Additional Data**

Following the reviewers' suggestions, we evaluated nine additional scenes from two diverse real-world datasets: five from ScanNet and all four indoor scenes from MipNeRF360. For MipNeRF360, we set the subset sample size to 20% instead of our standard 5% setting, as the original views are already sparse. We followed the official method for the MipNeRF360 test split, uniformly sampling one for every eight images. For the ScanNet test split, we randomly sampled 5% of the trajectory.

Observations from the real-world scenes align with findings from the Replica dataset in the main table. For most scenes, our sampling strategy with $z_{\text{DF}}$ demonstrates high novel view synthesis performance. This suggests that our method is robust in handling varying trajectories also in real-world scenes.

As mentioned in our article, the performance is related to the camera motion patterns in the data. For instance, in the ScanNet 0085_00 scene, the camera movement speed and position distribution are relatively uniform, which favors uniform sampling. The similar phenomenon can also be observed in the MipNeRF360 "counter" scene and the Replica dataset mentioned in the main text. However, non-expert users often deviate from such controlled camera motion patterns, which is precisely the motivation behind the methods proposed in this paper.

|                      | **0134_02**  | **0073_01**  | **0085_00**    | **0084_00**    | **0050_00** | **room**    | **counter** | **kitchen**  | **bonsai**|
|:---------------|:------------------|:-----------------|:------------------|:------------------|:------------------|:------------------|:-------------------|:-------------------|:-------------------|
| Random               | 27.84       | 25.7        | 28.17    | 16.7        | 21.66       | 26.17     | 24.56       | 25.34       | 27.15 |
| Uniform              | 29.26 | 27.94 | **28.34** | 16.7        | 21.66       | 26.72     | **25.03**   | 26.19       | 28.34 |
| $z_{\text{CF}}$  | 26.05       | 26.43       | 25.3     | 17.91| 22.79 | 27.21 | 24.95       | 26.48 | 27.78 |
| $z_{\text{DDP}}$ | 28.96       | 27.68       | 27.85    | 17.47       | 22.46       | 27.18     | 24.97       | 26.39       | 28.01 |
| $z_{\text{DF}}$  | **29.53**   | **28.16**   | 28.32 | **18.24**       | **23.03**   | **27.32** | 25.01 | **26.66**   | **29.32** |

---

### Decision · Action_Editor_icHn · 2025-04-22

**Recommendation:** Accept as is

**Comment:**

The reviewers all recommend acceptance, finding the paper well structured, the approach well motivated, and the experiments convincing. The review and rebuttal period appears to have been productive as well, making the paper even stronger. The AE recommends acceptance without further review necessary, but the authors are encouraged to integrate the work presented in the rebuttal.

**Audience:**

Indoor novel view synthesis is a high-impact problem, and theoretically-grounded tools for this area should be attractive to many.

**Claims And Evidence:**

The claims are properly supported by convincing evidence, especially with the additions provided in the rebuttal phase.